# A marine heatwave drives significant shifts in pelagic microbiology

Mark V. Brown [1,2 ✉], Martin Ostrowski [2], Lauren F. Messer [3], Anna Bramucci[2], Jodie van de Kamp [1], Matthew C. Smith[1], Andrew Bissett [1], Justin Seymour [2], Alistair J. Hobday [1] & Levente Bodrossy [1]

Marine heatwaves (MHWs) cause disruption to marine ecosystems, deleteriously impacting macroflora and fauna. However, effects on microorganisms are relatively unknown despite ocean temperature being a major determinant of assemblage structure. Using data from thousands of Southern Hemisphere samples, we reveal that during an "unprecedented" 2015/16 Tasman Sea MHW, temperatures approached or surpassed the upper thermal boundary of many endemic taxa. Temperate microbial assemblages underwent a profound transition to niche states aligned with sites over 1000 km equatorward, adapting to higher temperatures and lower nutrient conditions bought on by the MHW. MHW conditions also modulate seasonal patterns of microbial diversity and support novel assemblage compositions. The most significant affects of MHWs on microbial assemblages occurred during warmer months, when temperatures exceeded the upper climatological bounds. Trends in microbial response across several MHWs in different locations suggest these are emergent properties of temperate ocean warming, which may facilitate monitoring, prediction and adaptation efforts.

[1] CSIRO Environment, Hobart, Australia. [2] Climate Change Cluster, University of Technology Sydney, Ultimo, Australia. [3] Division of Biological and Environmental Sciences, University of Stirling, Stirling, Scotland. ✉email: oceanmicrobes@gmail.com

Marine heatwaves (MHWs) are prolonged anomalous ocean warming events that are emerging as consequential disruptors of coastal ecosystems globally[1,2]. While the drivers of individual MHWs can be region-specific and complex, most extreme MHWs are linked to phases of large-scale climate modes[3], some of which are increasing in frequency and intensity due to climate change[4,5]. Similarly, the frequency, intensity, and duration of MHWs has been increasing over the last century, linked to anthropogenic global warming[6,7] and this trend is projected to continue[1]. In fact, by the late 21st century widespread near-permanent MHW status could be the "new-normal" across large oceanic regions[8]. Indeed, reconstructed climatological datasets suggest many parts of the ocean are already experiencing a near constant state of heat stress compared to conditions a century ago[9].

Several well-documented extreme MHW events around the globe have had significant ecological and socioeconomic impacts. Ecological impacts include mass mortality events in marine fauna and flora such as fish[10], abalone[11], corals[12,13], seagrass[14] and kelp[15]. In addition to species range extensions, contractions, disruption to phenology[16] and local extinction due to thermal displacement[15,17], potential decreases in net primary productivity[18] and the development of harmful algal blooms[10] can lead to impacts on commercially important fisheries and aquaculture systems[19], or even total collapse[20]. The loss of biodiversity, ecosystem function and productivity from MHWs has knock on economic costs, often exceeding billions of dollars[21]. While severe consequences from MHWs have been documented for macroscopic assemblages, the response of microbial assemblages, which form the base of marine food-webs and perform important functional roles within marine ecosystems, remains relatively understudied[18,22–24]. Nevertheless, the relationship to temperature is consistently identified as a key trait determining microbial biogeography and assemblage composition[25–27] and thus extreme warming events are considered highly likely to impact microbial community structure and function[28].

In many macro-ecological studies observational data collected along oceanic gradients has been used to describe the organismal realised niche and define indicators such as the optimal temperature for a species occupation, the species temperature index (STI) and related community indicators, such as the community temperature index (CTI), which describes the abundance-weighted average thermal affinity of all the organisms present. These approaches have proven useful in describing and predicting species' biogeography[29] and providing insights into how whole ecosystems might respond to medium to long-term warming or cooling[30] and localised extreme events[31]. By constructing standardised frameworks, such metrics facilitate quantitative comparisons between data from different locations, times and environments regardless of species composition. From a microbial perspective, using in situ observations of the realised niche based on census data provide an assessment that accounts for biotic phenomena and synergistic stressors such as selective grazing, variability in viral mortality and microbial interactions that modulate the purely genetically defined "fundamental" growth optima generally captured in laboratory settings[32]. Indeed the "observed" oceanic niche may substantially differ from the "fundamental" niche[33]. Here, we collated a highly standardised molecular dataset describing Southern Hemisphere marine microbial composition in thousands of samples linked to in situ oceanic conditions (Supplementary Figs. 1 and 2 and Supplementary Table 1). Samples originate from latitudes 0–66 °S, depths 0 m–~6000 m and water temperatures −2–32 °C in the Pacific, Indian and Southern Oceans and the Tasman, Coral, Arafura and Timor Seas, spanning globally relevant gradients of light, temperature and nutrients. The combined molecular and oceanographic dataset was used to generate indices describing the generalised niche characteristics or environmental preferences of microbial species and assemblages. We use these indices to elucidate the impacts on pelagic microbiota of MHWs in temperate waters. Of particular focus was the 2015/16 Tasman Sea MHW which has been described as 'unprecedented' in its duration and intensity[34]. The event was captured during repeat sampling at the long-term Integrated Marine Observing System (IMOS) National Reference Station (NRS) at Maria Island in the Tasman Sea. Marine waters in this region have experienced pronounced warming at rates well above the global average[35]. Indeed, half of Australian coastal waters have experienced their warmest ever monthly temperatures since 2008[36]. Much of this warming has been associated with boundary currents such as the East Australian Current (EAC), which transport warm oligotrophic waters from the tropics into temperate latitudes and have been linked to profound ecosystem changes, including 'tropicalisation' of macrofauna and flora, as well as microbial assemblages[37]. We sought to determine if our niche-based framework could reveal previously undocumented impacts of this extreme warming event, and if so whether these impacts were greatest when temperatures exceeded the long-term climatological maxima.

## Results and discussion

**The microbial relationship to temperature.** Microbial STIs were calculated for ASVs representing an average of 97.9% of bacterial, 95.2% of archaeal and 93.4% of eukaryotic assemblage structures (note only STIs that were estimated in > 100 replicate analyses were included; Supplementary Table 2) allowing us to generate highly representative community level indices based on the combined STI of each ASV in the assemblage weighted by its relative abundance.

Overall, as expected, CTI was significantly aligned with environmental temperature (Fig. 1, Supplementary Fig. 3 and Supplementary Table 3) highlighting that at a global scale bacterial, archaeal and microbial eukaryotic filtering associated with organisms' thermal optima efficiently tracks marine temperature changes. Assemblages with a CTI above in situ temperatures (i.e. above the 1:1 line in Fig. 1) are described as under positive thermal bias, being composed of taxa preconditioned to higher temperatures, and so may display reduced sensitivity to oceanic warming. Conversely, assemblages under negative thermal bias (below the 1:1 line in Fig. 1) are composed of taxa with thermal optima lower than in situ conditions and so may be more sensitive to warming. The CTI ~ temperature relationship tends to flatten at temperature extremes (e.g. slope estimate from linear regression for voyage samples <10 °C = 0.705, 10 °C to 20 °C = 0.98, >20 °C = 0.486 and similarly, at the southern most NRS station MAI = 0.378, at the mid-latitude station PHB = 0.767 and at the northern most station DAR = 0.241; see parameter estimates for linear regression, Supplementary Table 3), but particularly for bacteria in the warm range (Fig. 1), such that assemblages in waters warmer than ~25 °C are under negative thermal bias while those waters cooler than ~5 °C are under positive thermal bias. This pattern is repeatable using other independent global microbial datasets (Supplementary Fig. 4) and is also consistent with global observations of macro-organisms such fish and invertebrates[29]. Similarly, waters between 100 and 1000 m are generally under negative thermal bias while those in waters >1000 m are under positive thermal bias (Fig. 1a). Such broad global patterns are thought to result from the fact that very few species have abundance optima near the edge of their thermal range[38]. As range is fundamentally constrained by the maximum and minimum ocean temperatures, species living near these extremes

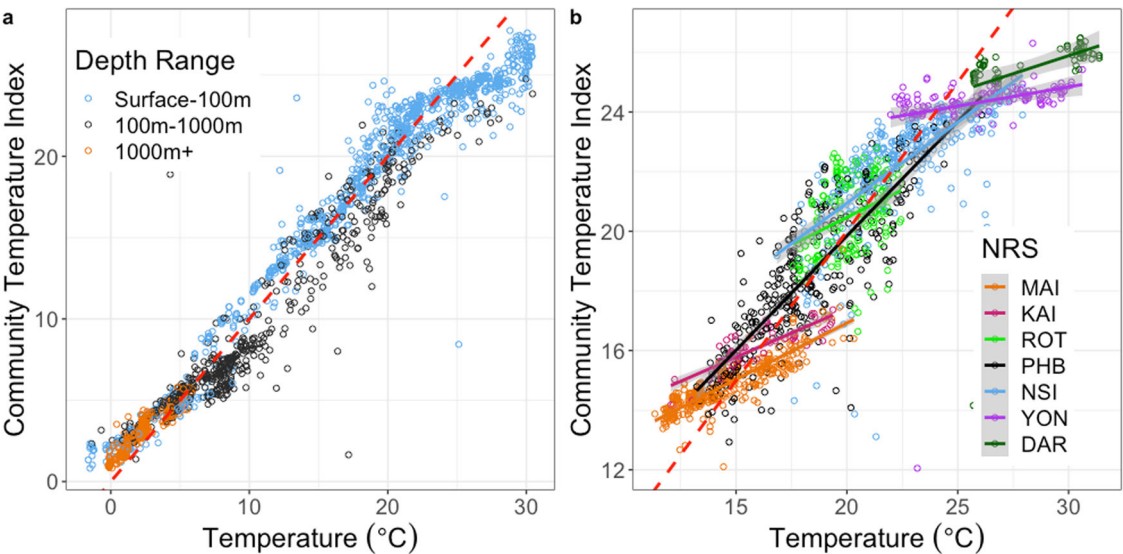

**Fig. 1 The relationship between bacterial community temperature index and in situ environmental temperature. a** Samples collected spatially during oceanic voyages in the Southern Hemisphere ($n = 1946$). **b** Samples collected temporally at IMOS National Reference Stations (NRS) time-series sites around the Australian continental shelf ($n = 1660$). Dashed red line represents a slope of one.

tend to have optima slightly warmer (in the poles) or slightly cooler (in the tropics) than these absolutes, resulting in combined community compositions that display either positive or negative thermal bias. Assemblages that are already under thermal stress, i.e. displaying negative thermal bias, are likely to be where the greatest ecosystem perturbations from oceanic warming due to global climate change will occur. Importantly, within this overall envelope, the CTI ~ temperature relationship varied considerably when sampled over time at different locations. At the seven NRS sites, while uniformly positive, the slope of the microbial regression ranged from almost flat at stations in the tropics (DAR, YON) to nearer 1:1 in temperate systems (PHB, ROT, NSI) and flatter again in the cool temperate waters (MAI, KAI) (Fig. 1b and Supplementary Table 3). Depending on where each slope crossed the 1:1 line these relationships resulted in systematic patterns of seasonal thermal bias at a local level (Supplementary Fig. 5). We next sought to examine how localised warming during MHWs affects microbial assemblages against this backdrop of seasonal thermal bias.

**Microbial thermal response to marine heatwaves.** The MAI NRS is located in the temperate Tasman Sea. Environmental conditions here are highly seasonal (Fig. 2), with water temperatures peaking in late Austral summer/ early autumn (February–March) and troughs occurring in late winter/early spring (August–September). Over the temporal sampling period, from February 2012–2020, this site experienced 34 MHWs (Supplementary Table 4), most of which lasted less than two weeks and were thus too ephemeral for our sampling regime to capture. The most sustained and intense event, however, occurred during the Austral summer of 2015/16 as part of a widespread Tasman Sea MHW event[34,39] (Fig. 2).

Anomalous poleward advection of warm water into the Tasman Sea by the East Australian Current (EAC; Australia's Pacific western boundary current) and its eddies is considered to be the dominant contributing factor to the "unprecedented" 2015/16 MHW event in the Tasman Sea[34,39]. The resultant MHW was mainly localised along the southeast Australian coast (Fig. 2a and Supplementary Fig. 6a–c). Sea surface temperatures were the warmest ever recorded for that region, with anomalies

up to 3 °C being observed over thousands of square kilometres[34]. The event lasted for many months (from 6 September 2015 — 9 May 2016). At the MAI NRS this event manifested as two prolonged MHWs interspersed with ~1 week of non-MHW conditions (Maria Island MHW No's 68 and 69: Supplementary Table 4). Combined, these MHW events lasted 241 days (excluding the 7-day interval), had a peak intensity of 3.69 °C and represented a cumulative intensity (sum of all heatwave days and the average heatwave intensity in °C) of 499.3. Because the definition of MHW events used here is based on satellite data, and so restricted to surface waters, deeper waters may not have undergone unusual warming during the periods we identified. Indeed, during the 2015/16 MHW, while the upper water column was in a state of MHW, below ~50 m the water was unusually cool, and actually cooled until mid-February 2016 (Fig. 2b). However, waters rapidly warmed in late February leading to warm temperature anomalies for deeper waters during late summer and autumn 2016[34] (Fig. 2b). A transition to very low nutrient concentrations occurred at the same time, suggesting a sustained deep-water intrusion over the continental shelf gave way to warmer oligotrophic poleward flowing EAC derived waters very rapidly in late February 2016.

Based on CTI ~ temperature relationship at the MAI NRS, bacterial and archaeal assemblages are under some degree of seasonal thermal stress during the months November–April when waters are warming to their peak (Supplementary Fig. 5a). However, while they display a similar seasonal cycle, eukaryote assemblages maintain a higher CTI and remain under positive thermal bias throughout most of the year (Supplementary Fig. 5a). We generated linear models describing the CTI ~ temperature relationship at MAI for each depth and kingdom separately. This enabled the identification of samples that deviated from long-term trends (for example, where the CTI was higher or lower than would be expected for a given in situ temperature, indicating times when there is significant deviation from the normal seasonal mechanisms driving community assembly). In total, across the three kingdoms (bacteria, archaea and eukaryote) we identified 53 significant outliers (>four times mean Cook's Distance; Supplementary Fig. 7), 72% of which occurred in samples collected during MHWs and 51% of which occurred during the 'unprecedented' 2015/16 Tasman Sea MHW (Oliver

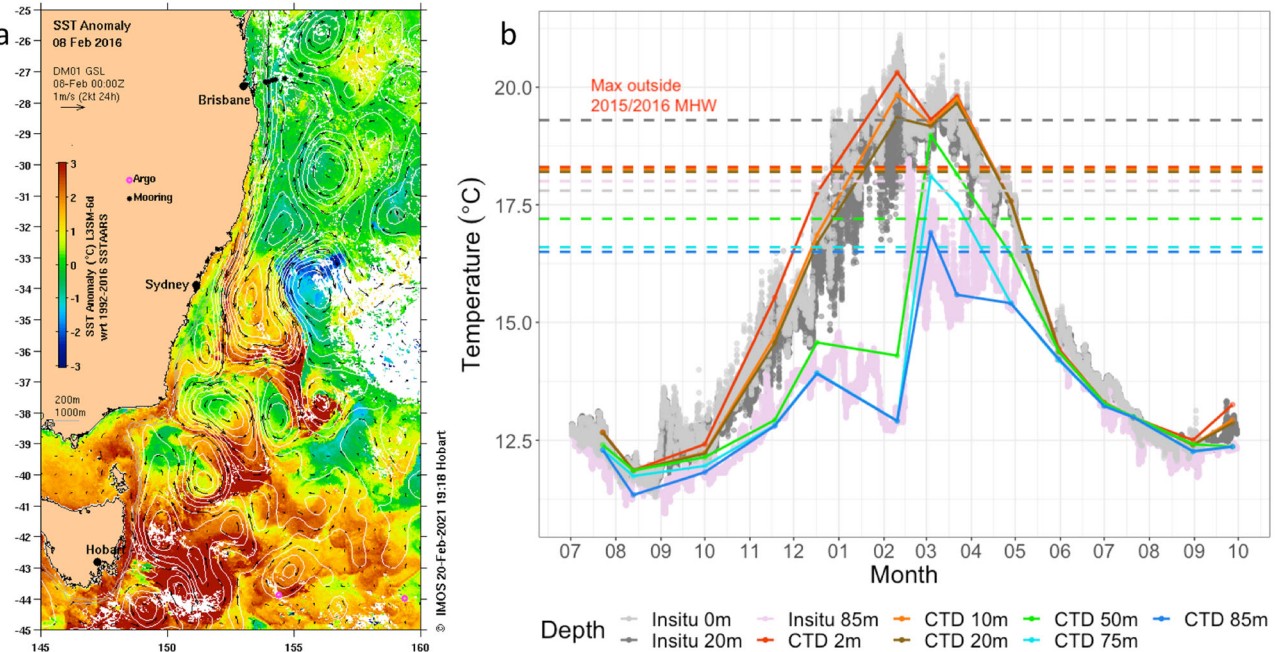

**Fig. 2 Conditions during the 2015/16 marine heatwave at Maria Island. a** Map displaying SST anomaly during the 2015/16 MHW event in the Tasman Sea. Six-day composite night-only SST anomaly and current velocity images are centred around February 8th, 2016; date of peak intensity of 2015/16 MHW at Maria Island National Reference Station (42° 35.80 S, 148° 14.00E). Data was sourced from Australia's Integrated Marine Observing System (IMOS) – IMOS is enabled by the National Collaborative Research Infrastructure Strategy (NCRIS). It is operated by a consortium of institutions as an unincorporated joint venture, with the University of Tasmania as Lead Agent. **b** Temperature profile of the Maria Island National Reference Station water column from July 2015 through October 2016, capturing the MHW event. Data consists of in situ monitoring at depths 0 m, 20 m and 85 m, as well as ship-based CTD profiling at depths 2 m,10 m, 20 m, 50 m, 75 m and 85 m collected at the time of sampling for molecular analysis. Horizontal lines represent the highest temperature at each depth to be observed outside the 2015/16 MHW event.

et al 2017). The highest water temperature recorded during the sampling period outside of summertime MHW conditions was 18.3 °C. When MHW activity forced in situ temperatures above this level, the slope of the bacterial and archaeal CTI ~ temperature relationship increased (Supplementary Fig. 7a, b), suggesting selection pressure based on temperature was stronger when these unprecedented temperatures were reached. The eukaryote assemblage maintained a positive thermal bias until higher temperatures, with the CTI ~ temperature slope crossing the 1:1 line at ~17.5 °C, nearer the peak MHW temperatures (Supplementary Fig. 7c). This suggests eukaryotic microbes experienced less temperature stress during MHW conditions. Even so, the highest CTIs observed at Maria Island for all three kingdoms and for all six depths (except Archaea at 50, 75 and 85 m) occurred near the peak of the 2015/16 MHW, during sampling on either the 4th or the 22nd of February 2016, after several months of the near continuous MHW activity (Fig. 3a and Supplementary Figs. 8a and 9a). The cold summer conditions in deeper waters, followed by warm temperature anomalies, are also evident in the signatures of the microbial assemblages. While peak CTI in deeper waters occurred in February, the CTI during November–January sampling periods remained at comparatively low values for summer periods (Fig. 3a).

Temperature based selection can be observed by directly comparing the STI of taxa that were present in greater or lesser abundances during the 2015/16 MHW event than in equivalent months during non-MHW conditions (Fig. 3b). The STI of taxa that were positively selected for (i.e. had a greater relative abundance during MHW conditions) in surface waters (0–10 m) effectively tracked the increasing environmental temperatures as the MHW progressed, resulting in a thermal bias for these organisms centred on zero (Fig. 3b and Supplementary Figs. 8b

and 9b). Conversely, taxa that were selected against displayed negative thermal bias and, for many bacteria and archaea, in situ MHW temperatures exceeded even the upper temperature of their normal range (Tmax: upper quartile of temperature profile) (Fig. 3c and Supplementary Fig. 8c), especially during the peak MHW months of February and March. In deeper waters taxa selected both for and against generally tracked environmental temperatures during all months until the March sampling period (Supplementary Figs. 10–12), which was when deeper water temperatures peaked and entered MHW status (Fig. 2b). Ultimately, the 2015/16 MHW event drove environmental temperatures to approach or exceed the upper limits of the known thermal distribution of many taxa normally inhabiting the Maria Island NRS.

**The niche state of pelagic microbial assemblages.** We extended our temperature-based framework to include other environmental variables widely available for our dataset (Supplementary Fig. 2 and Supplementary Table 1), resulting in the generation of species and community salinity (SSI/CSI), nitrate + nitrite (SNI/CNI), phosphate (SPI/SPI), silicate (SSiI/CSiI) and oxygen (SOI/COI) indices. As with temperature, at a global scale most community indices reflect efficient microbial filtering along environmental ocean gradients (Supplementary Table 5) and at MAI reflected the temporal dynamics of seasonal cycles and associated effects of water column mixing and stratification. CSI, CNI and CPI for each kingdom were positively correlated (Pearson correlation coefficient > 0.5) with in situ environmental conditions (Supplementary Table 5). Additionally, the archaeal CSiI was correlated with in situ silicate concentrations (Pearson correlation coefficient = 0.76) and bacterial and eukaryotic COI were

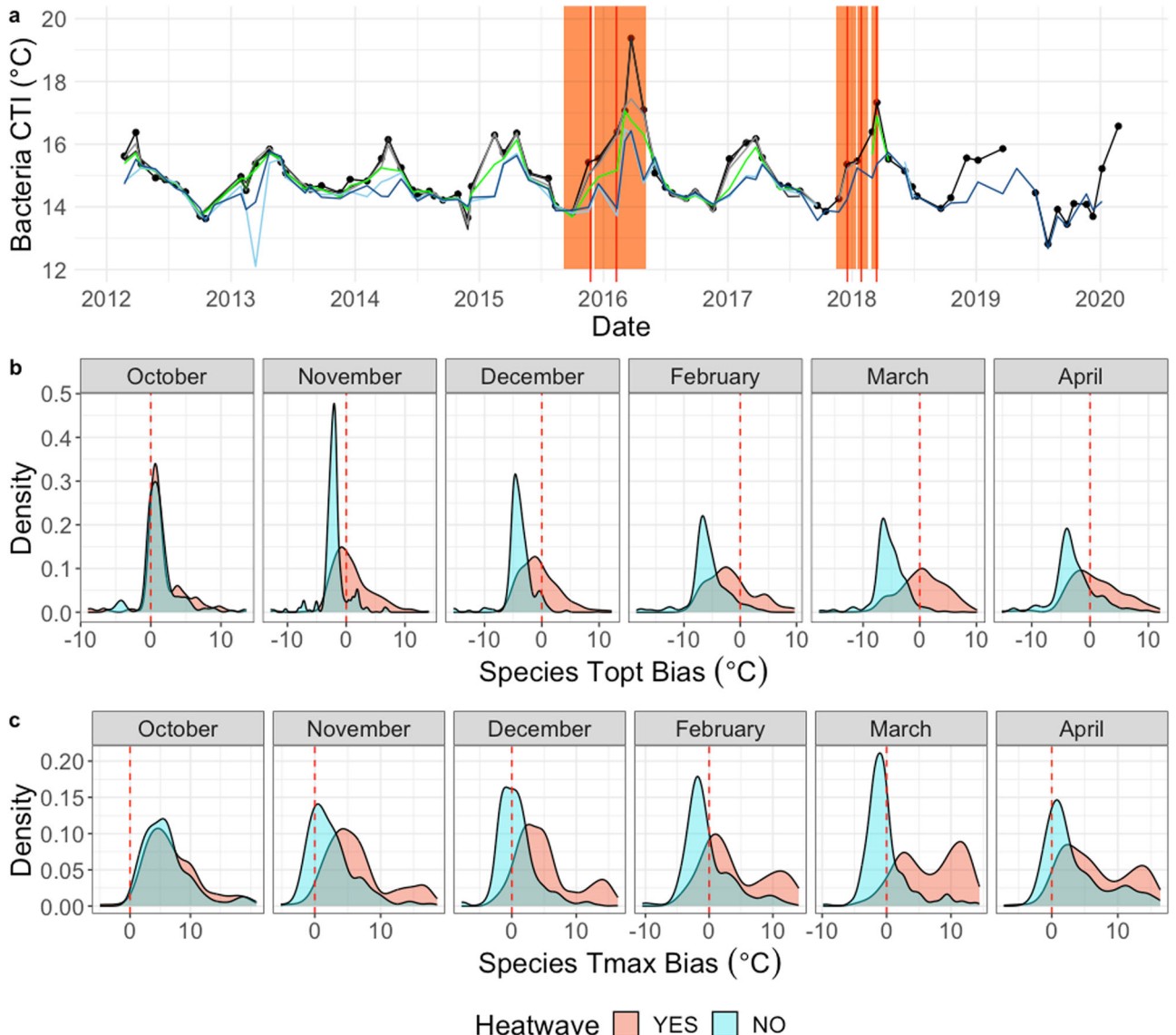

**Fig. 3 The evolution of temperature selection for bacteria during the 2015/16 Tasman Sea MHW. a** Bacterial community temperature index (CTI) at the Maria Island NRS during the sampling period. Lines correspond to depths: Surface (black), 10 m (dark grey), 20 m (light grey), 40 m (dark green), 50 m (light green), 75 m (light blue) and 85 m (dark blue). The 2015/16 and 2017/18 heatwave periods are shown as light red background, with peak intensity for each MHW event identified with a red bar. **b** Density plots display the distribution of the thermal optima and (**c**) the thermal maxima of bacteria selected for (total n across all months =2017) or against (total n across all months =1735) in surface waters (0 and 10 m depth) during the 2015/16 MHW, compared to equivalent months during non-heatwave conditions (YES=selected for during the heatwave event, NO=selected against during the heatwave event). Dashed red line indicates zero bias. Notably, during February and March, the thermal maxima of many organisms selected against during the MHW is exceeded.

correlated with in situ oxygen concentrations (Pearson correlation coefficient 0.51 and 0.73 respectively; Supplementary Table 5). As with temperature, the 2015/16 MHW resulted in extreme levels for most of these indices, with the highest CSI and lowest CNI, CPI, and COI observed at MAI all occurring during this event (Supplementary Figs. 13–17). Additionally, this MHW accounted for 27% of all CSI outliers, 50% of all CPI outliers and 38% of all CNI outliers in the trend analysis (Supplementary Figs. 18–20).

Another series of MHWs occurred in the Tasman Sea during the Austral summer of 2017/18. From 16 November 2017 through 19 March 2018 at Maria Island three heatwave periods occurred that rendered MAI under MHW status for 110 days out of the 124-day period. While the 2015/16 event was localised to Australia's eastern seaboard, the 2017/18 event spread further east, covering the entire Tasman Sea (Supplementary Fig. 6 compares SST anomalies in the Maria Island region between A-C) 2015/16 MHW and D-F) 2017/18 MHW), with SST anomalies reaching 3.7 °C in the eastern Tasman Sea near New Zealand[16]. Rather than advective heat associated with the EAC extension, it has been suggested the 2017/18 event was more closely tied with local air–sea heat fluxes[40] and coincided with significant land-based heatwaves in both Tasmania and New Zealand[16]. Although sampling of this event was not as comprehensive, and the cumulative intensity of the heatwave at MAI was not as intense (245.52 vs. 499.30 in total for the 2017/18 and the 2015/16 heatwaves, respectively), we did observe similar general trends in community indices, especially in surface samples. Temperature and salinity-based indices of all kingdoms increased to near the peak levels observed during the 2015/16 MHW, along with low

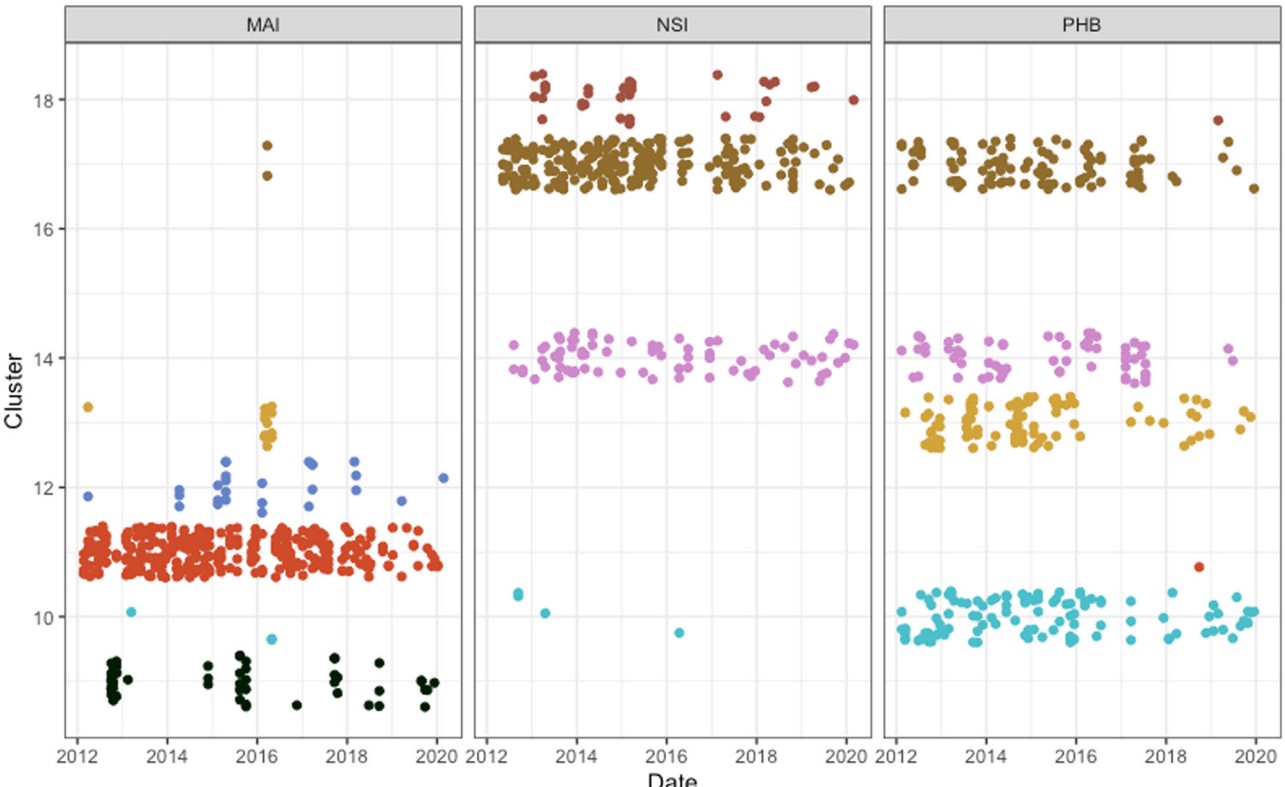

**Fig. 4 Cluster state of bacterial assemblages at National Reference Stations Maria Island (MAI), Port Hacking (PHB) and North Stradbroke Island (NSI).** During the heatwave in 2015/16 assemblages at Maria Island transitioned to niche states generally observed at stations situated hundreds of kilometres to the north (equatorward). (Legend: Black = cluster 9, Turquoise = 10, Red = 11, Blue = 12, Yellow = 13, Pink = 14, Brown = 17, Maroon = 18).

levels in nutrient-based indices (Fig. 3a, Supplementary Figs. 8a, 9a, and 13–17), illustrating a repeatable pattern of selection for community niche characteristics under heatwave conditions. Further, given the different proximate causes of each of these MHW events, the observation of similar trends suggests they are intrinsically linked to the ocean warming itself, rather than poleward advection of microbial assemblages (via the EAC, which has been linked to shifting microbial compositions at sites further north[37]).

**The environmental niche status of microbial assemblages shifted to novel states during the 2015/16 MHW in the Tasman Sea.** We clustered samples based on the values of their community indices, providing a single reference cluster for each sample in each kingdom (bacteria n = 20 clusters, archaea n = 20, eukaryote n = 14) reflecting the generalised niche characteristics or environmental preferences of the assemblage (Supplementary Figs. 21–23). Clusters are indexed based on increasing mean of the community temperature index. Over basin scales these clusters resolve along lines of depth and latitude, providing insight into oceanic zones where biological niche transitions occur (Supplementary Fig. 24), while resolving over temporal scales at the local level. Bacterial samples collected at MAI NRS generally resolved into clusters 9, 11, and 12, with 77% (329/427) falling within cluster 11 (Fig. 4). When transitions between cluster states are observed, they are aligned with seasonal changes. For example, 49 out of 52 samples classified as cluster 9 occur in samples collected between August and November, corresponding to the late Austral winter and spring (Supplementary Fig. 25). During this time waters are cooler and strong winds result in a well-mixed water-column (Fig. 2b). Additionally, in mid- to late summer, when the water column becomes stratified, another

transition to cluster 12 occurs, mostly in surface samples (Supplementary Fig. 25). However, during the 2015/16 MHW, samples from MAI transitioned to clusters 13 and 17 (Fig. 4 and Supplementary Fig. 25). These two clusters represent communities with quite different niche characteristics to those usually observed at MAI NRS, with affinities for higher temperature and salinities and lower nutrient and oxygen requirements (Fig. 5). For both clusters 13 and 17, the MAI NRS, at latitude 42° S, represents the furthest south (poleward) they were observed in our entire dataset (Supplementary Fig. 24; mean/maximum (equatorward) latitude for cluster 13 = 35.1° S/29.9° S and cluster 17 = 27.3° S/10.5° S) and they are rarely, if ever, observed at MAI outside of this MHW event (cluster 13 was recorded once in March 2012). Indeed, these niche state clusters are generally characteristic of microbial assemblages inhabiting warm oligotrophic water masses much further to the north (equatorward), including at National Reference Stations PHB (latitude 34° S), NSI (27° S) (Fig. 4) and YON in the Great Barrier Reef Lagoon (19° S; Supplementary Fig. 24). Cluster 17 in particular is often observed in the Pacific western boundary current (WBC) the East Australian Current and as far north as 10.5° S in the Coral Sea, 1936 nautical miles (3585 km) to the north of MAI (Supplementary Fig. 24).

**Marine heatwaves modulate the diversity and structure of microbial assemblages.** The move towards higher community temperature and lower nutrient indices (Fig. 5) is a result of the niche preferences of the organisms selected for and against during MHW conditions. Not all "winners" under MHW conditions result in taxonomic turnover. For example some ubiquitous taxa such as the SAR11 clade Ia are composed of many strains that resolve along temperature gradients[26] and, in relation to MHWs,

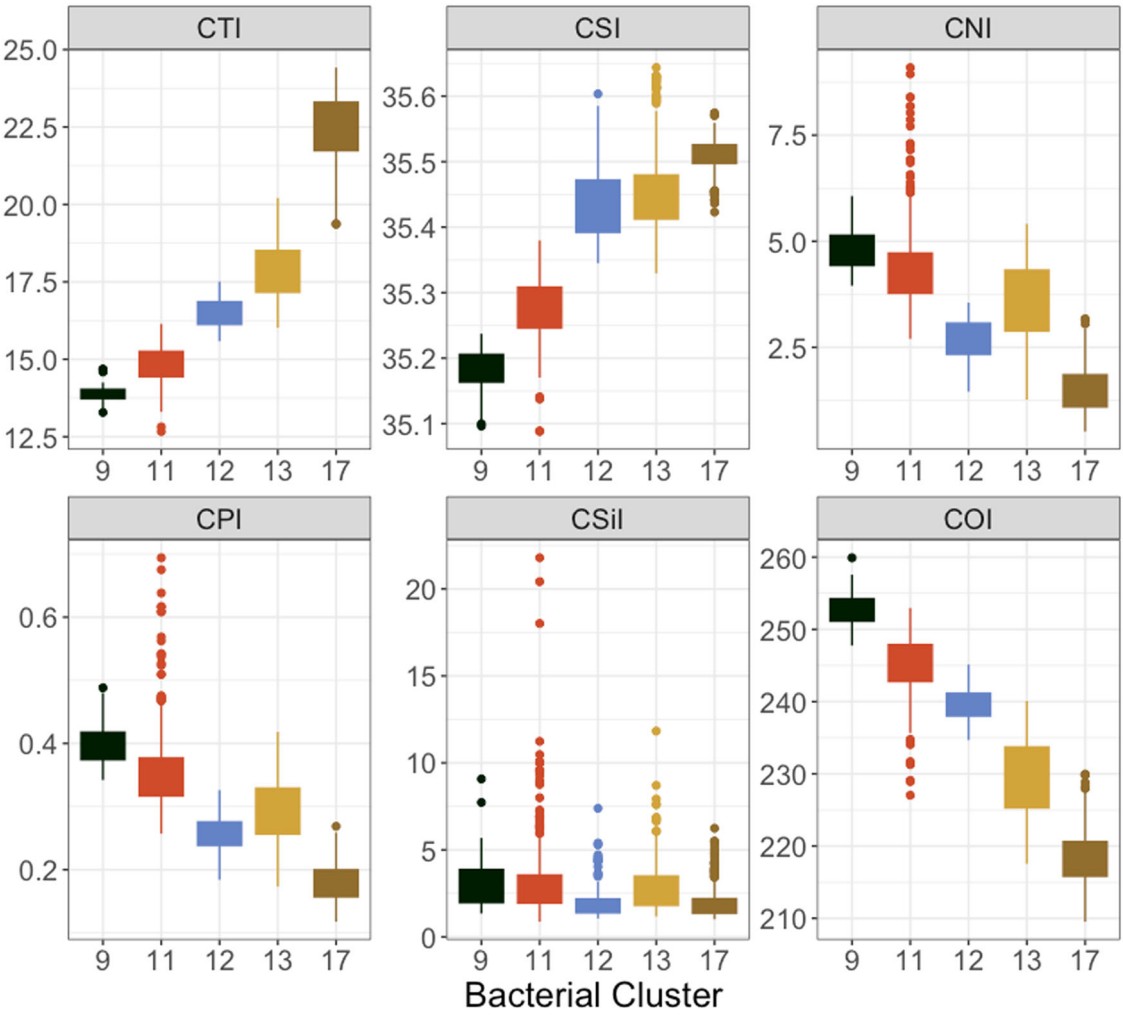

**Fig. 5 Relationship between bacterial niche-based clusters observed at Maria Island and community indices.** CTI community temperature index, CSI community salinity index, CNI community NOx index, CPI community phosphate index, CSiI community silicate index, COI community oxygen index.

transitions between strains adapted for different temperatures occur (mean STI of SAR11 Ia strains selected for/against each month of the MHW; October 14.54 °C/12.87 °C; November 17.11 °C/13.87 °C; December 18.20 °C/14.04 °C; February 17.05 °C/14.93 °C; March 19.91 °C/14.33 °C; April 19.70 °C/15.75 °C). However, many of the taxa that were favoured during MHW conditions display well established preferences for warm, oligotrophic conditions, including the marine cyanobacteria (*Synechococcus*, *Prochlorococcus* and the diazotrophic *Trichodesmium*), photoheterotrophic bacteria such as the SAR86, SAR116 (Fig. 6a) as well as photoheterotrophic Archaeal lineages MGIIb-O1, MGIIb-O3, MGIIb-O5, and MGIIa-L1, (Supplementary Fig. 26a) which grow particularly well in warm (median/optimum temperatures > 20 C) low nutrient waters[41,42]. Within the eukaryotes, MHW conditions favoured many largely uncharacterised dinoflagellates clades (Supplementary Figs. 27a and 35) as well as small unicellular algae (Chlorophytes) of the Class Mamiellophyceae (as identified using chloroplast 16S rRNA gene sequence data; Supplementary Fig. 28). Conversely, MHW conditions saw a transition away from bacterial groups such as Flavobacteria clades NS2b, NS5 and NS7 and the Rhodospirillales AEGEAN-169 Marine Group (Fig. 6a) that are often associated with particle attachment to larger phytoplankton.

Similar transitions away from large phytoplankton and their associated bacteria, towards unicellular phototrophs (such as the marine cyanobacteria and Chlorophytes) and photoheterotrophic

bacteria and archaea have been observed in warming eastern Australian waters linked to the seasonal transport by the East Australian Current[37]. Interestingly, these transitions also align with those observed during a large and sustained MHW in the northeast Pacific Ocean during 2014/15, (see[43] and reference therein). At both the northerly and southerly extent of this MHW (northerly station Ocean Station Papa (OSP), 50 ° N, 145 ° W, in the oceanic subarctic Pacific;[22,23] southerly stations in the Southern California Current (SCC) ecosystem at 29.84 − 37.61 °N, 117.28 − 125.75 °W[24]), marine cyanobacteria, particularly *Prochlorococcus, Synechococcus* increased in relevance as waters became more oligotrophic. Further, at OSP photoheterotrophs including SAR11 Ia and the MGII Archaea were favoured and unicellular Chlorophytes also proliferated, while at SCC an array of dinoflagellate taxa were favoured.

Taken together, these common responses (i.e. MHW phytoplankton assemblages dominated by small, unicellular Chlorophytes, dinoflagellates and cyanobacteria, along with increased numbers of photoheterotrophs), may be fundamental results of warming in temperate marine waters. Importantly, these shifts can have profound ecosystem consequences as the rates, magnitude and fate of carbon fixed by small unicellular phytoplankton can be quite different to that fixed by larger organisms such as diatoms[37].

Non-metric multi-dimensional scaling (nMDS) analysis shows surface samples from MAI during MHWs form distinct grouping

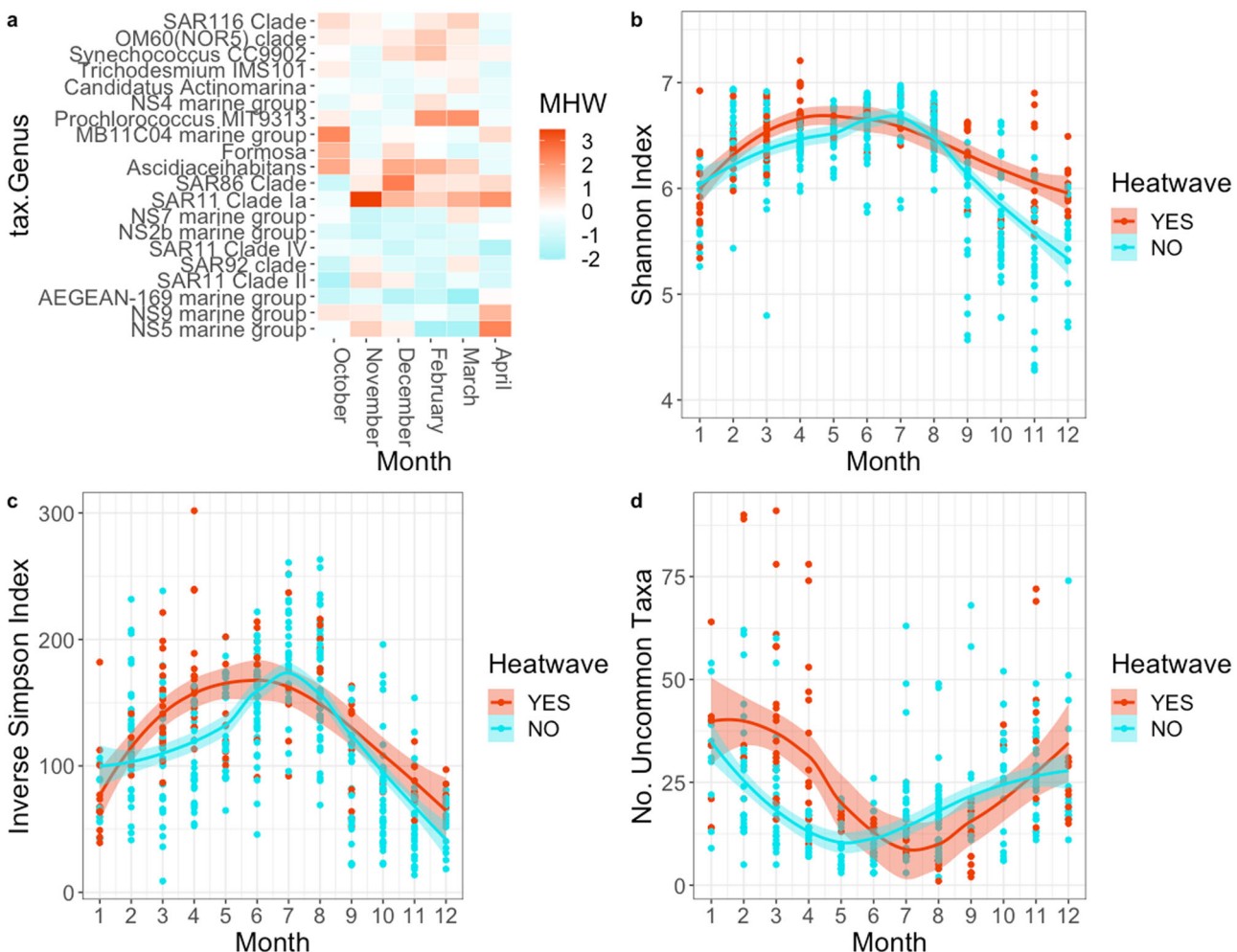

**Fig. 6 Marine heatwaves lead to compositional and structural changes in microbial assemblages. a** Heatmap detailing the bacterial genera contributing most to the compositional difference between surface samples collected in equivalent months during MHW and non-MHW conditions. **b** The seasonal cycle of bacterial Shannon diversity, (**c**) Inverse Simpsons diversity and (**d**) the number of "uncommon" bacterial taxa in each sample are all modulated by MHW condition. Samples collected during MHW conditions ($n = 121$) are in red and those collected during non-MHW conditions ($n = 378$) are in blue.

to those observed at other times at MAI or at PHB (Supplementary Fig. 29). That is, while niche preferences of the MHW assemblages may align to those observed in assemblages farther north (Fig. 4), the phylogenetic composition and relative abundances of taxa remain distinct. To determine how the community structure of these assemblages may be altered we further examined aspects of diversity and composition.

MAI displays a strong seasonal cycle of microbial diversity, a pattern that has previously been linked to high environmental heterogeneity at the site[44]. During non-MHW conditions both Shannon and Inverse Simpson measures of diversity peak during Austral winter months although the timing of the peak differs slightly between kingdoms (Bacteria peak between June -August, Archaea around July-September and Eukaryotes peak between May-July (Fig. 6b, c and Supplementary Figs. 26 and 27). When MHW conditions occur during the warmest months of March and April, bacterial and archaeal diversity actually increases significantly (Fig. 6b-c; Supplementary Fig. 26b, c and Supplementary Tables 6a, b and 6a, b) compared to levels during non-MHW conditions, and in the case of bacteria modulates the shape of the seasonal diversity cycle by shifting the peak towards spring. Indeed the highest bacterial diversity levels observed occurred during MHW conditions. Bacterial and eukaryotic diversity also increase during MHW activity in spring and early summer

(November and December), when diversity typically reaches its lowest levels, leading to a further moderation in the degree of variability in the seasonal cycle (Fig. 6b, c, Supplementary Fig. 27 and Supplementary Tables 6 and 8).

The increases in diversity during MHWs, along with the appearance during the 2015/16 MHW of unusual taxa such as *Prochlorococcus* and *Trichodesmium erythreum* (which overall occurred in <1% of samples from the cool temperate MAI NRS), suggest heatwave conditions support the appearance of uncommon taxa at this site. An important consequence of ocean warming on macro-organisms has been the proliferation of species redistributions and range shifts, leading to the formation of assemblages that facilitate novel biological interactions[45]. This phenomenon has been well documented around the Australian coastline, including in our study region[46]. Given that many of the important ecosystem services provided by microorganisms are a result of biological interactions[47], we sought to consider to what level the introduction of new species into the environment was a driver of diversity and restructuring in microbial assemblages. To do this we identified how often each ASV occurred at MAI over the sampling period, based on presence/absence. ASVs were defined as common if they occurred in > 90% of samples and uncommon if they occurred in < 10% of samples. For the bacteria, out of 15,674 total ASVs, this resulted in a "core" group of 140

common ASVs. Most ASVs ($n = 11,978$) were uncommon but the majority of these also displayed low overall relative abundances, forming part of the well-documented rare tail of microbial assemblages[48,49]. To determine if uncommon taxa were becoming more prevalent during MHWs than expected, we examined the top 500 most abundant ASVs in each sample (representing the top ~1/3 of ASVs, with the average number of unique ASVs per sample being 1,511) thus avoiding the rare tail where sampling may not reveal robust patterns. Overall, bacterial assemblages during MHWs contained a significantly greater proportion of uncommon (but relatively abundant) taxa when they occurred during the late Austral summer, with particularly high values between February – April, Fig. 6d and Supplementary Table 6C), times when diversity is high and the microbial assemblage is already in a state of thermal deficit or stress (Supplementary Fig. 5). Conversely, MHW samples also contained fewer of the common taxa, however they generally retained well over 100 (of the 140) core bacterial ASVs. Thus, when MHWs occurred during warmer months, forcing local temperatures beyond upper levels of the long-term climatology, they supported relatively high levels of uncommon, warm adapted bacteria while retaining many core species, leading to increased diversity and the development of novel assemblage structures, as seen in macro-organisms. Although these assemblages may only persist in the short term, if, as expected under future climate scenarios, near continuous MHW conditions are achieved in these temperate regions[8], the local pelagic microbiota will undoubtedly undergo significant change, with concomitant effects on the ecosystem services they provide.

## Conclusions

Here we develop and demonstrate a framework for examining the impacts of MHWs (and longer-term climate-related warming) on microorganisms that is aligned with efforts developed for macro-organisms. Because these indices are fundamentally linked to assemblage composition they provide a metric of turnover along with a direction of selective pressure. While direct comparisons of molecular methods to data generated by other methods (e.g. microscopy) at the species level should be undertaken with caution (see Methods and Supplementary Data 2), comparison between the magnitude and directional response of community level indices across taxonomic boundaries (e.g. microbes to phytoplankton to zooplankton to fish) will likely prove fruitful as generalised whole of ecosystem indicators. Our indices are available to be visualised and interrogated using the IMOS Biological Ocean Observer application[50] (shiny.csiro.au/BioOceanObserver/) While focusing here on extreme temperature events, our approach lends itself to examining the effects of other ecosystem perturbations such as cold spells, storms that cause extreme rainfall, turbulence and mixing etc., where selection is likely to occur along complex gradients of environmental traits. These indices provides a way to distil the complex microbial response to extreme marine events into simple metrics that could be used as a warning or alert signal to facilitate clear, effective communication and aid public comprehension[51]. Using this framework, we address a considerable deficiency in knowledge concerning MHW impacts on the coastal ocean. We show that extreme warming events, such as the 2015/16 MHW in the Tasman Sea, have the capacity to drive profound transitions in the niche characteristics of microbial assemblages and modulate strong seasonal cycles of diversity. Common trends in indicators across multiple heatwaves, as well as common taxonomic responses observed at other locations, suggest that transitions to small, unicellular phototrophs and photoheterotrophs may be emergent properties of temperate ocean warming. Such shifts can have cascading effects across the food chain and will provide a focus for monitoring, prediction and adaptation efforts moving forward[52].

## Methods

**Marine heatwave definitions**. Heatwave events, defined as periods of at least five consecutive days when daily sea-surface temperatures (SSTs) exceed the 90th percentile of climatological (seasonal) SST observations (calculated between 1/1/1982 and 31/12/2011)[53] were identified at Australia's Integrated Marine Observing System (IMOS) Maria Island National Reference Station (NRS) using coordinates −42.625 °S, 148.375 °E in the "marineheatwaves.org" heatwave tracker tool. This tool uses the daily Optimally Interpolated Sea Surface Temperature (OISST) data from the National Oceanic and Atmospheric Administration (NOAA), specifically the AVHRR-only v2.0 data from 1982 to 2015 and the AVHRR-only v2.1 data from 2016 to present day. All events identified at Maria Island during the sampling period are outlined in Supplementary Table 4.

**Ocean sampling and molecular analysis and bioinformatics**. All data were collected and analysed in a standardised manner in accordance with the protocols of the Australian Microbiome (see the Ausmicrobiome Scientific Manual; https://confluence.csiro.au/display/ASM). Samples from oceanographic transects in the Pacific, Indian and Southern Oceans and the Tasman, Coral, Arafura and Timor Seas span latitudes $0 - 66$ °S, depths from surface to ~6,000 m, (Supplementary Fig. 1) and water temperatures −2 to 32 °C. Temporal samples are included from seven near decadal Integrated Marine Observing System (IMOS) National Reference Stations (NRS)[54]. These sites, located at Maria Island (MAI; 42° 35.80 S, 148° 14.00 E), Kangaroo Island (KAI; 35° 49.93 S, 136° 26.84 E), Rottnest Island (ROT; 32° 00.00 S, 115° 25.00 E), Port Hacking (PHB; 34° 05.00 S, 151° 15.00 E), North Stradbroke Island (27° 20.50 S, 153° 33.73 E), Yongala (19° 18.51 S, 147° 37.10 E) and Darwin Harbour (DAR; 12° 24.00 S, 130° 46.08 E) span ~30° of latitude along the Australian continental shelf, occupying temperate, subtropical and tropical waters with minimum and maximum sea surface temperatures (SST) ranging between 11 °C and 31 °C. They are positioned to maximise spatial representation of Australian marine bioregions[55].

For each sample, 2 litres of seawater was collected and filtered without pre-processing through a 0.2 μm filter. DNA was extracted and purified using the DNeasy® PowerWater® Sterivex™ DNA Isolation Kit (Qiagen, Germany). Samples collected at National Reference Stations underwent an additional Phenol:-Chloroform:Isoamyl Alcohol step[54]. The quality and quantity of DNA was checked using a NanoDrop™ 8000 Spectrophotometer (Thermo Scientific™). Bacterial, archaeal and microbial eukaryote partial small subunit ribosomal RNA genes were amplified independently, but from the same extracted DNA sample using primer sets bacterial 27f (AGAGTTTGATCMTGGCTCAG[56]) and 519R (GWATTACCGCGGCKGCTG[57]); archaeal A2F/Arch21f (TTCCGGTTGATCCYGCCGGA[58]) and 519 R* (GWATTACCGCGGCKGCT[56]) and eukaryote TAR-euk454FWD1 (CCAGCASCYGCGGTAATTCC[59]) and TAReuk-Rev3 (ACTTTCGTTCTTGATYRATGA[60]). Sequencing was carried out using either 250-bp (eukaryote 18S rRNA gene) or 300-bp (bacterial and archaeal 16S rRNA gene) paired-end sequencing on the Illumina platform at the Ramaciotti Centre for Genomics (University of New South Wales, Sydney). All environmental metadata including physical (temperature, salinity, oxygen), biological (chlorophyll) and nutrient parameters (nitrate/nitrite, phosphate, silicate) were collected and analysed

following the standard procedures of the Integrated Marine Observing System (IMOS)[61].

The open-source program "R" was used for amplicon read processing, statistical analysis, and production of figures[62]. All maps and graphs were generated using the package "ggplot2" unless otherwise stated. Map boundaries were provided by "rnaturalearth" libraries. Illumina paired R1 and R2 reads were processed using the DADA2 pipeline (version 1.2.0;[63]), adapted slightly to ensure the highest number of merged reads after processing (https://github.com/martinostrowski/dada2.pipeline). Sequence data was processed using the DADA2 package to identify unique amplicon sequence variants (ASVs). Briefly, reads with 'N' bases were discarded and bacterial V1-V3 primers were truncated using "cutadapt"[64], R1 and R2 were trimmed to remove low-quality terminal ends, then denoised, merged, and chimeras were removed using the dada2 removeBimeraDenovo program at a less stringent minFoldParentOverAbundance=4 (advised for use when expecting a higher rate of less divergent distinct amplicon sequence variants (ASVs) within the dataset). Quality control measures resulted in the removal of ~50% of raw sequence reads. Initial and final read numbers are provided in Supplementary Data 1 and histograms of final read numbers are presented in Supplementary Fig. 30. Mean number of sequences per sample in final tables were 54484 in the bacterial table, 81231 in the archaeal table and 76985 in the eukaryotic table.

Bacterial and archaeal 16S rRNA ASVs were taxonomically identified using the Genome Taxonomy Database Tk-v 1.5.0 (GTDB;https://gtdb.ecogenomic.org/), chloroplast 16S rRNA ASVs and Eukaryote 18S rRNA ASVs were identified using the PR2 databases (version 5.0.1;[65]). The quality ASV tables were then secondarily filtered to remove all ASVs not annotated to the target kingdom. Samples required > 5000 total reads to be included in downstream analyses.

Model-based clustering and classification of assemblages based on community indices was performed using the 'mclust5' package[66]. Standardised (zero mean, unit standard deviation) community weighted index data were pre-processed using the Uniform Manifold Approximation and Projection (UMAP; version 0.5) algorithm[67] for dimensional reduction. The number of components (clusters) in each finite Gaussian mixture model (GMM) was based on examination of Bayesian Information Criterion (BIC) and integrated complete-data likelihood criterion (ICL).

**Statistical and reproducibility.** Shannons diversity (H) was calculated as $-\sum_{i=1}^{Sobs} \frac{n_i}{N} \ln \frac{n_i}{N}$ and Inverse Simpsons (I/D) as $1/\sum_{i=1}^{Sobs} \frac{ni(ni-1)}{N(N-1)}$ where $S_{obs}$ = number of unique ASVs in a sample, $n_i$ = relative abundance of the ith ASV, and N = total abundance of ASVs in the sample.

For non-metric multidimensional scaling (nMDS) and Similarity Percentage (SIMPER) analyses, observations of zero abundance were replaced in ASV tables with a non-zero estimate value using the count zero multiplicative (CZM) function from the zCompositions[68] package, and centred log ratio (CLR) transformation was carried out using the CoDaSeq package[69]. Resultant tables were imported with environmental data into the Primer VX software where nMDS and SIMPER were performed to identify ASVs differentially abundant between MHW and non-MHW conditions in equivalent months. Taxonomic heatmaps displaying differentially abundant taxa were generated using Simper outputs for each month aggregated to Genus (Bacteria, Archaea) or Class (Eukaryote, Chloroplasts) taxonomic level and scaled by subtracting the mean and dividing by the standard deviation using the scale function in R with default settings.

Linear models describing the relationship between community weighted indices and in situ environmental variables at each sample depth and outlier detection of those samples with a Cook's distance four times the mean of all points, were carried out using the lm() and cooks.distance() functions.

For each ASV we calculate the optima and range of its relationship with available environmental parameters including temperature, salinity, nitrate/nitrite, phosphate, silicate, oxygen and chlorophyll (see https://confluence.csiro.au/display/ASM/meth_6.1.1+Microbial+Niche+Indices).

Relative abundance data along with associated environmental variables were fit to a Kernel Density Estimation (KDE) function using the density() function in the stats package. This is a non-parametric smoothing technique used to estimate the probability density function of a random variable. The value of the environmental variable at the maximum point of the KD curve (point of maximal estimated abundance), was defined as the species optima. Species ranges were calculated as the difference between the minimum and maximum of the environmental parameter where kernel density = ¼ peak height. We allowed the normal distribution of the kernel density model to continue 3 °C each side of the minimum and maximum temperature in each instance to allow for potentially negatively or positively skewed range dynamics (e.g. organisms with temperature optima close to or at the minimum or maximum range). Examples of raw data and KD models are provided in Supplementary Fig. 31 and Supplementary Table 10. To test robustness of the kernel density approach we also calculated temperature optima based on the mean temperature of the samples in which each ASV displayed its four highest abundances in the dataset. These two methods were highly aligned (linear model of comparison: Bacteria slope = 0.97, $r^2 = 0.998$; Archaea slope = 0.98, $r^2 = 0.998$; Eukaryote slope = 1.04, $r^2 = 0.995$). For each index we report the mean and standard deviation based on 1000 repeat KDE calculations. To remove sampling bias, for each of the 1000 iterations, samples were selected randomly with replacement from bins along the variable gradient, based on the number of samples in the smallest sized bin. Each ASV was required to be present in at least 100 samples in the subset for the calculation to be performed. In all we calculated 6 indices and associated measures, which are illustrated by the example of Temperature below.

Species Temperature Index (STI) is a measure of the optimal temperature of the realised thermal niche of an organism (represented by an ASV) for a given environmental variable. It was calculated as (1) the value corresponding to the peak of the abundance-weighted KD model (Supplementary Fig. 31) and (2) using the mean of the value of the variable for samples containing the top 4 relative abundances. The unit of selection in marine microbial assemblages is often at the sub-species "eco-type" or clade level[70,71] which requires the highest possible molecular resolution to identify, hence the use of single nucleotide resolved ASVs. Despite this we retain the nomenclature "species" temperature index etc. in order to align with similar efforts in maro-organisms and to facilitate comprehension by non-specialists.

Species thermal range (STR) is the temperature range for a defined species abundance, here calculated as the difference between the minimum and maximum temperatures where kernel density = ¼ peak height (Supplementary Fig. 31).

Species thermal bias (T-bias, STI - $ST_{insitu}$) is the difference between STI and environmental sea temperature at the point of sampling (ST) and thus is not a static factor for any given species. Thermal bias can also be calculated using the species maximum temperature (Tmax; maximum temperatures where kernel density = ¼ peak height) rather than optimum (STI), providing an indication of periods where environmental temperatures

approach or exceed the upper limits of a species known thermal distribution.

Community temperature index (CTI, $\sum_{i=1}^{i=N} STI_i w_i / \sum w_i$) is the average thermal affinity of the entire assemblage. Calculated as the sum of the realised thermal optimum (STI) of each organism (i) present weighted by their relative abundance ($w_i$), divided by total organismal abundance in the sample. Only STIs with > 100 estimates were used to calculate CTI values.

Community thermal bias (T_bias, CTI - ST$_{insitu}$) is the difference between CTI and environmental sea temperature (ST) at the point of sampling.

To test the effect of upstream bioinformatics choices on the KD approach to community weighted index generation, we compared the results detailed here with those that are now output on an ongoing basis by the Australian Microbiome (AM). The AM generates the community weighted indices from the same sample set using the same code as described here, but, for historical reasons, uses different intermediate bioinfomatic protocols (see the AM methods manual for details https://confluence.csiro.au/display/ASM/meth_5.1+Amplicon+Analysis). The major differences between protocols are that the AM utilises the USEARCH v11 algorithm[72] to resolve ASVs (as opposed to DADA2 herein) and species and community indices are generated using samples rarefied to 20,000 reads (as opposed to un-rarefied sample with >5000 reads here). Because the KD approach relativises the input data as part of the code, these protocol changes result in almost negligible difference in outcome (Supplementary Table 9). For example, linear model parameters correlating CTI estimates (using the KD approach) from here against the AM estimates for Bacteria (Estimate = 1.006, Std. Error = 0.0007, $t$-value = 1427.81, $r2 = 0.998$, F-statistic = 2.039e + 06(1,3182), $p < 2.2$e-16), Archaea (Estimate = 0.998, Std. Error = 0.002, $t$-value = 594.87, $r2 = 0.992$, F-statistic = 3.539e + 05(1,3034), $p < 2.2$e-16) and Eukaryotes (Estimate = 0.995, Std. Error = 0.002, $t$-value = 652.57, $r2 = 0.994$, F-statistic = 4.258e + 05(1,2436), $p < 2.2$e-16) are all highly significant.

While independent verification of all indices is beyond the scope of this work, we provide working examples to highlight how our index dataset recapitulates well established baseline patterns of critical taxa in marine environments generated for each kingdom while supporting exploration and identification of novel findings. Examples describe niche characteristics of clades within the marine cyanobacteria (Supplementary Fig. 32), the Marine Group II Archaea (Supplementary Fig. 33) and the unicellular green algae of the Class Mamiellophyceae within the Eukaryotes (Supplementary Fig. 34). Finally, to highlight the discovery aspect of the data we present the niche parameters of largely uncharacterised and uncultivated dinoflagellates in clades I-IV within the Syndiniales, a ubiquitous group of endoparasites (Supplementary Fig. 35). Our results highlight how these clades are structured along oceanic gradients, with some displaying very tight ecological constraints that could be used to track environmental change.

We also provide a comparisons of the STI derived here with those estimated for some phytoplankton species that are common along the eastern Australian coast as estimated by light microscopy based on whole water and continuous plankton recorder dataset[73] (Supplementary Data 2). These comparison highlight both synergies and discrepancies between the approaches and suggest caution should be employed when directly comparing the two (Supplementary Data 2).

**Use of TARA Oceans datasets**. Using the same kernel density code as described in the methods, we used publicly available miTag data (16S rRNA gene sequence tag abundances) from 0.8 − 0.2um filtered metagenomic samples, downloaded from the Tara Ocean companion website (http://ocean-microbiome.embl.de/companion.html) as abundance values along with TARA oceans metadata to produce community temperature indices for each TARA metagenomic sample, as an independent test of the global patterns in temperature bias observed using the AM data (Supplementary Fig. 4). Additionally, again using the same code, we used abundance of mapped reads from 236 TARA oceans metagenomes against 204 de-replicated Archaeal Marine Group II (MGII) metagenome-assembled genomes (MAGs; see[42] for details) along with TARA oceans metadata to derive species indices for each MGII MAG. We display comparison between AM amplicon derived MGII indices and MAG derived MGII indices for phylogenetic clades[42]. Shapiro-Wilk normality test was used to assess normality of untransformed or log transformed data before Pearson correlation coefficient was used to determine correlation between clade niche parameters derived using AM and TARA datasets.

**Examples of baseline data**. To establish the robustness of our approach, we provide examples of how using KD to define niche parameters recapitulates established patterns of niche characteristics within each kingdom.

*The marine cyanobacteria*. Marine cyanobacteria are globally distributed and account for ~25% of ocean net primary productivity, their success a result of the proliferation of ecotypes adapted to the range of oceanic gradients. Our indices recapitulate patterns of clade niche differentiation based on thermal and nutrient adaptions that have been well-documented previously. For instance, Prochlorococcus display optima > 15 °C as they are essentially absent at temperatures below this[71]. Prochlorococcus high light-adapted clades (HL-I and HL-II) have niche optima centred around low nutrient and high oxygen conditions found in marine surface waters, while low light clades (LL-I, LL-II/III) reside deeper in the water column where nutrient are higher and oxygen levels lower. Interestingly our data suggest a potential relationship between LL-II clade and silicon levels that has not to our knowledge been reported previously. HL-I displays broader temperature and nutrient niche characteristics than HL-II[33,74], although occupies a narrower range of salinities. HL-II clades generally have a higher thermal optima then Hl-I clades[33,71] and the mean STI observed in our data aligns well with the fundamental niche temperature optima reported by Smith et al.[33] (HL-I = 22.3 °C, our data 22.2 °C; HL-II = 24.16 °C, our data = 25.1 °C.) Similarly, Synechococcus Clades I/IV (mean STI = 15.2 °C) which co-dominate polar and temperate waters are adapted to lower temperatures and higher nutrient levels than Clade II/III (mean STI = 23.8 °C), which inhabit warm oligotrophic water[75–77] (Supplementary Fig. 32). Our data also reveal differentiation between Trichodesmium clade I and III along temperature (mean STI Clade I = 26.3 °C, Clade III = 23.8 °C) and phosphate (mean SPI Clade I = 0.035 μmol/L, Clade III = 0.151 μmol/L) gradients as suggested by Rouco et al.[78].

*The Marine Group II Archaea*. Planktonic marine Archaea generally fall into three main groups, the Thaumarchaeota (Marine Group I), Marine Group II (MGII) and Marine Group III. MGII are ubiquitous, largely photoheterotrophic organisms inhabiting photic zone (surface to mesopelagic) waters around the globe. Rinke et al.[42] used 270 MAGs to phylogenetically describe 21 genus level MGII clades. We used our KD approach with the values of read mapping 236 TARA oceans metagenomes[27] against these MAGS as abundance data along with TARA ocean metadata to estimate niche characteristics for these clades and

compare to the values obtained using our AM amplicon dataset. While the data are totally independent and produced in different ways there are many commonalities in the results. Although not all clades were observed in our dataset (missing deep-sea clades MBIIa-J1, MBIIa-J2, MBIIa-J3 and MGIIb-Q2, surface clades MBIIa-K2, and MGIIa-L4 which was restricted in TARA data to Arabian Sea) both reveal MGII genus level niche partitioning along temperature (Pearson's correlation between AM and TARA mean clade niche estimates $r(13) = 0.51$, $p < 0.05$), nitrogen ($r(13) = 0.68$, $p < 0.01$), phosphate ($r(13) = 0.48$, $p < 0.005$), silicate($r(13) = 0.56$, $p < 0.05$), oxygen ($r(13) = 0.64$, $p < 0.05$). Again, the ubiquity of these organisms in the global surface and mesopelagic ocean is facilitated by adaptation to the wide environmental gradients with clear distinctions between those adapted to warm, oligotrophic waters (e.g. MGIIb-N1, MGIIb-N2, Poseidonia sp., MGIIa-L3,) and those that inhabit cooler temperature and broader nutrient regimes found in deeper waters and temperate/polar oceans (e.g. MG2b-Q1 MGIIb-O2, MGIIb-P, MGIIb-L1, MGIIa-I)

*The marine Mamiellophyceae.* Within the Eukaryotes the Family Mamiellophyceae (unicellular green algae) are abundant and widespread oceanic primary producers which, like the marine cyanobacteria, have undergone a radiation of distinct phylogenetic clades adapted to different ecological gradients[79]. Again our data accurately resolves the environmental niche characteristics of these clades as recently described based on Ocean Sampling Day datasets and historical analyses[79,80]. Briefly, *Bathycoccus prasinos* displays generally broad niche characteristics as all known cryptic species within this complex contain identical 18S rRNA gene sequences and are indistinguishable using this marker. The genus *Ostreococcus* is absent from very cold polar waters. *O. lucimarinus* is restricted to cool-temperature (mean STI = 15.0 °C) mesotrophic waters but is absent from tropical waters, while *Ostreococcus sp.* Clade B inhabits warm oligotrophic subtropical and tropical environments (mean STI 19.5 °C). *Mantoniella squamata* (mean STI = 18.2 °C) and *Mantoniella sp.* clade B (mean STI = 25.3 °C) niche characteristics separate along temperature and salinity gradients. The genus *Micromonas* separates in 9 clades defined by temperature optima, ranging from polar and sub-polar clades *M. polaris* (mean STI = 0.06 °C) and *Micromonas bravo* Clade B3 (mean STI = 4.25 °C) throughout temperate conditions (*M. commode* A2 mean STI = 12.7 °C, *M. pusilla* mean STI = 14.6 °C, *Micromonas bravo* B1 mean STI = 15.1 °C, *Micromonas bravo* B2 mean STI = 17.8) and into subtropical and tropical environments (*Micromonas commoda* A1 mean STI = 24.4 °C, *Micromonas bravo* B5 mean STI = 29.7 °C).

*Comparison of CTI to those derived using continuous plankton recorder.* While these examples provide evidence that the our approach generates suitable baseline data for important marine microorganisms, very few similar attempts to define species or community level environmental indices for marine microbes exist against which to benchmark our data. One of particular note, Ajani et al.[73], employed the kernel density approach to resolve species temperature indices for 30 phytoplankton species commonly identifies along the east coast of Australia[73]. They used phytoplankton abundance data harvested by light microscopy identification from samples collected at IMOS NRS ( ~ 750 samples[81]) and by the Australian Continuous Plankton recorder survey (~6000 samples[82]). They provide tables detailing 30 phytoplankton species and their STI (Supplementary Table 1 in Ajani et al.[73] as well a list of common phytoplankton not included un their analysis for various reason (Supplementary Table 2; Ajani et al.[73]). We have populated these tables, reporting results for all ASVs in our dataset that were taxonomically

identified to the species level that match each phytoplankton, including 12 of the 30 species for which STI were calculated, and for a further 38 species that were not analysed (Supplementary Data 2). Where more than one ASV was identified at the species level we also provide mean and standard deviation for the collective. This provides the capacity to be built upon as more data becomes available, as well as to see where there may be cryptic species resolvable by our molecular markers that inhabit different oceanic niches, or where molecular annotation databases may need updating. Of the phytoplankton represented by only one or a few molecular ASVs, some display STI's very similar to those calculated from microscopy work (e.g. *Tripos fusus* STI = 17.2 °C/ Ajani et al.[73] STI = 17.1 °C; *Meuniera membranace*a 21.5 °C/ 19.9 °C) while others are quite different (e.g. *Asterionellopsis glacialis* 18.7 °C/12.9 °C; *Phalacroma rotundatum* 24.8 °C/ 18.9 °C). Similarly, where taxa are represented by many ASVs the mean of ASV STIs may equate well with the microscopy based estimate (e.g. *Cylindrotheca closterium* 20.1/21.9 °C) while others may be quite divergent, however some phytoplankton with multiple ASVs are clearly composed of cryptic strains, such as *Proboscia alata* which separates into a polar strain with an STI = 2.2 °C, and a temperate/tropical strain with STI 23.3 °C which compares well with Ajani et al.[73] STI = 22.7 °C (Supplementary Data 2).

**Reporting summary**. Further information on research design is available in the Nature Portfolio Reporting Summary linked to this article.

## Data availability

Sequence data from the Australian Microbiome is available under NCBI Bioproject number PRJNA385736. Additionally, ASV count tables and metadata are available using the Australian Microbiome Initiative Bioplatforms Data Portal (https://data.bioplatforms.com/bpa/out). All species and community weighted indices and diversity indices reported here are available through the IMOS Biological Ocean Observer (shiny.csiro.au/BioOceanObserver/) and the Australian Microbiome Github (https://github.com/AusMicrobiome/microbial_ocean_atlas). All environmental metadata including physical (temperature, salinity, oxygen), biological (chlorophyll), and nutrient parameter are available via the Australian Ocean Data Network Portal (https://portal.aodn.org.au/). In situ temperature, salinity, oxygen, turbidity, chlorophyll Water Quality Meter data for 20 m and 85 m at Maria Island (https://portal.aodn.org.au/search?uuid=8964658c-6ee1-4015-9bae-2937dfcc6ab9). Near real-time meteorology and sea surface temperature at Maria Island (https://portal.aodn.org.au/search?uuid=f3910f5c-c568-4af0-b773-13c0e57ada6b). Discrete depth temperature, salinity, carbon, nitrate, phosphate, silicate, oxygen, chlorophyll for National Reference Stations (https://portal.aodn.org.au/search?uuid=b442c3e8-3d30-48ad-b144-680afd848167). All historical satellite data products are available using IMOS Ocean Current (https://oceancurrent.aodn.org.au/).

## Code availability

All RCode required for the calculation of microbial species and generalised community weighted indices using the kernel density and mean abundance approach from a 3 column species x abundance x environmental-variable table is publicly available on the Australian Microbiome Github (https://github.com/AusMicrobiome/microbial_ocean_atlas).

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

## Acknowledgements

We would like to acknowledge the contribution of the Marine Microbes (MM) and Biomes of Australian Soil Environments (BASE) projects, through the Australian Microbiome Initiative in the generation of data used in this publication. The Australian Microbiome Initiative is supported by funding from Bioplatforms Australia through the Australian Government National Collaborative Research Infrastructure Strategy (NCRIS). Data were sourced from Australia's Integrated Marine Observing System (IMOS) – IMOS is enabled by the National Collaborative Research Infrastructure (NCRIS). Data were also sourced through the Marine National Facility and we thank the officers, crew and researchers involved during cruises of the *R/V Southern Surveyor* (ss2010_v09, ss2012_t07, ss2013_t03), the *R/V Aurora Australis* (AA2014/15_v2, AA2015/16_v3) and the *R/V Investigator* (IN2014_e03, IN2015_c02, IN2015_v03, IN2016_t02, IN2016_v03, IN2016_v04, IN2017_v01, IN2019_v03).

## Author contributions

M.V.B., A.H., and L.B. defined the research question for this study. M.V.B., M.O., L.F.M., A.B., J.v.d.K., M.S., A.B., J.S., and L.B. funded the work, collected samples, performed laboratory work, performed bioinformatics and statistical analysis, and interpreted the results. M.B. drafted the manuscript. All authors contributed to the final manuscript.

## Competing interests

The authors declare no competing interests.
