## [Peer Review File · Communications Biology]

Reviewers' comments:

Reviewer #1 (Remarks to the Author):

In the study, the authors developed a suite of indices describing species and community level niche parameters using thousands of Southern Hemisphere marine microbial samples. This is an important study to understand the temperate microbial assemblages in response to marine heatwaves. The theme in this study is interesting and the analysis is well written. However, some suggestions (major revision) below that need to be considered to improve the quality of manuscript.

Line 22: What is the link between Marine heat waves and these environmental factors such as lower nutrient conditions. Clarify here.

Line 23: This sentence does not have relation with microorganism - you need to clearly summaries your results and their importance here.

Line 76: Please be more specific and add some points to discuss your experimental design. If possible, you need to state the hypothesis and/or aim of the study here.

Line 90: Some of the content seems to belong in the Materials and methods section rather than the results section.

Line 122: Statistical tests and values are not fully reported in these results, such as varied CTI ~ temperature relationship.

Line 348: These results don't seem to show up in pictures or tables.

Line 388: Additional evidence is needed to support these conclusions.

Line 452: The authors do not report the sequence of the sample range, and prove that the depth of the sample sequencing is sufficient. Whether the microbial data is flattened or not, the result of relative abundance may affect the establishment of the model. If using the rarefied samples is needed to improve the robustness of the findings?

Line 468: Whether this treatment that observations of zero abundance were replaced in ASV tables with a non-zero estimate value is reasonable, because the general ASV table usually has very high zero values.

Line 481: The authors calculated the optima and range of relationship of each ASV with available environmental parameters, however, whether there are false positives in low abundance, especially in ASVs converted from zero values. Whether it is more ecologically meaningful to retain only high abundance or core microorganisms for analysis.

Line 508: Due to the accuracy of annotation and sequencing, different ASVs may represent the same species, which is different from the index constructed by plankton or other marine organisms at a more precise level of species classification. And whether the applicability in species results annotated by 16s or 18s sequencing is verified.

Line 538: If possible, making the code available would help to assess the statistical rigor and validity of the analysis. Furthermore, this will allow future scientists to build upon the similar work.

Reviewer #2 (Remarks to the Author):

The COMMSBIO-23-2150-T study describes how a marine heat wave causes significant shifts in pelagic microbiology. The topic is highly relevant and timely as climate change threatens marine ecosystems and organisms worldwide. Marine heatwaves (MHWs) are prolonged anomalous ocean warming events that have attracted attention as triggers of, for example, coral bleaching events. Linked to global warming, MHWs have been increasing in frequency, intensity, and duration. Yet, most of the research to date has focused on oceanography or macroorganisms. Investigations into the relationship between microorganisms and climate change have long been hampered by comprehensive data sets and methodological limitations. Furthermore, most marine microbial time series that are able to detect

trends over time are located in the northern hemisphere, leaving huge knowledge gaps in the southern ocean regions. The present study not only uses an impressive dataset of thousands of samples to fill the knowledge gaps on the impact of MHWs on marine microbes in the Southern Hemisphere, but also generates novel indices and important insights that are transferable to other ocean regions, extreme events and groups of organisms. I therefore think that this paper is highly relevant and of interest for the microbial community and a wider field as well. The sequences and environmental data were found in the respective databases and the methods were described in a comprehensive way such that the data analysis should be reproducible.

Despite the overall great impression, I have a few comments that should be addressed:

I am confused by two sets of supplementary figures and tables (Extended data figures and Supplementary figures). I recommend combining these files into one document.

Fig2 - legends and formatting, the figure resolution in the provided pdf was bad. Please provide a better resolution and larger captions and legends.

Fig3 - please provide axis captions

Extended Data Figure 7. This figure is very striking and describing overall biogeographic community/niche patterns. Together with my below comment on bacteria community composition (L346ff) this would make an excellent overview figure.

L346ff - which taxa have the specific niches and can adapt most to the heatwave -> extend the presentation of results and discussions related to the identity of "winners" and "losers", also did alpha diversity or community evenness differ during MHW event and between bacteria, archaea, and eukaryotes? Please include a figure representing the community composition of communities in the different clusters or possibly in combination with Fig.3.

L360 - It would be interesting to see if/how the authors think that the study relates to the Metabolic Theory of Ecology (MTE) or emergent principles regarding increasing temperatures - see for example the recent review by Gralka 2023 <https://doi.org/10.1093/icb/icad060>.

L378 - "introduction of new species into the environment" Where would the species be introduced from if not transported with currents or from the rare biosphere?

L388-392 - is this trend statistically significant?

L446 and elsewhere, refer to the full amplicon identifier 16S rRNA instead of just "16S". Which primers/gene regions were used?

L467 was one combined ASV table across stations/projects generated for downstream analyses? How many samples did not pass the filter thresholds? A table with sequencing results and #sequences passing the bioinformatic filters should be added to the supplementary files.

L490 - It would be useful for the less experienced reader if the authors could show some examples of calculated species ranges as supplementary figure, e.g. of the species that were found to be least/best adapted to higher temperatures.

L507f - Please add the mathematical formulas for each index.

We thank both reviewers for taking the time to review our submission and feel they highlighted many aspects that, upon revision, make our work substantially better.

Reviewers' comments:

Reviewer #1 (Remarks to the Author):

In the study, the authors developed a suite of indices describing species and community level niche parameters using thousands of Southern Hemisphere marine microbial samples. This is an important study to understand the temperate microbial assemblages in response to marine heatwaves. The theme in this study is interesting and the analysis is well written. However, some suggestions (major revision) below that need to be considered to improve the quality of manuscript.

The first two queries concern wording in the Abstract which is character limited. We have done our best to rewrite the abstract to clarify the reviewers concerns

Line 22: What is the link between Marine heat waves and these environmental factors such as lower nutrient conditions. Clarify here.

We have changed the wording in the abstract to clarify that MHW drives the changes in environmental conditions. This sentence starting line 19) now reads

“Temperate microbial assemblages underwent a profound transition to niche states aligned with sites over 1000km equatorward, adapting to higher temperatures and lower nutrient conditions brought on by the MHW.”

Line 23: This sentence does not have relation with microorganism - you need to clearly summaries your results and their importance here.

This sentence has been removed. We have changed the wording in the abstract so it more accurately reflects the results of MHW effects on microorganisms, including altered niche states, modulation of seasonal patterns in diversity and the support for uncommon taxa.

Line 76: Please be more specific and add some points to discuss your experimental design. If possible, you need to state the hypothesis and/or aim of the study here.

We agree the introduction ended abruptly and have added the following text in response to the reviewers request

“Here, we collated a highly standardised molecular dataset describing Southern Hemisphere marine microbial composition in thousands of samples linked to *in situ* oceanic conditions (Supplementary Figure 1, 2, Supplementary Table 1). Samples originate from latitudes 0 - 66 °S, depths 0m - ~6,000m and water temperatures -2 - 32 °C in the Pacific, Indian and Southern Oceans and the Tasman, Coral, Arafura and

Timor Seas, spanning globally relevant gradients of light, temperature and nutrients. The combined molecular and oceanographic dataset was used to generate indices describing the generalised niche characteristics or environmental preferences of microbial species and assemblages. We use these indices to elucidate the impacts on pelagic microbiota of MHWs in temperate waters. Of particular focus was the 2015/16 Tasman Sea MHW which has been described as ‘unprecedented’ in its duration and intensity (Oliver et al 2017). The event was captured during repeat sampling at the long-term Integrated Marine Observing System (IMOS) National Reference Station (NRS) at Maria Island in the Tasman Sea. Marine waters in this region have experienced pronounced warming at rates well above the global average (Hobday and Pecl, 2014). Indeed, half of Australian coastal waters have experienced their warmest ever monthly temperatures since 2008 (Babcock et al 2019). Much of this warming has been associated with boundary currents such as the East Australian Current (EAC), which transport warm oligotrophic waters from the tropics into temperate latitudes and have been linked to profound ecosystem changes, including ‘tropicalisation’ of macrofauna and flora, as well as microbial assemblages (Messer et al 2020). We sought to determine if our niche based framework could reveal previously undocumented impacts of this extreme warming event, and if so whether these impacts were greatest when temperatures exceeded the long-term climatological maxima.”

Line 90: Some of the content seems to belong in the Materials and methods section rather than the results section.

We agree and have removed the following portion of this paragraph and integrated this information in a suitable manner into the methods section or elsewhere.

“Indices were calculated using a highly standardised molecular dataset describing Southern Hemisphere marine microbial assemblage composition (bacterial n=3644, archaeal n=3114 and eukaryotic n=2549) linked to *in situ* oceanic conditions (Supplementary Figure 1, Supplementary Table 1). Samples from oceanographic transects in the Pacific, Indian and Southern Oceans and the Tasman, Coral, Arafura and Timor Seas span latitudes 0 - 66 °S, depths from surface to ~6,000m, (Extended Data Figure 1) and water temperatures -2 to 32 °C. Temporal samples are included from seven near decadal Integrated Marine Observing System (IMOS) National Reference Stations (NRS) (see Brown et al 2018 for details). (Extended Data Figure 1). These sites, located at Maria Island (MAI; 42° 35.80 S, 148° 14.00 E), Kangaroo Island (KAI; 35° 49.93 S, 136° 26.84 E), Rottnest Island (ROT; 32° 00.00 S, 115° 25.00 E), Port Hacking (PHB; 34° 05.00 S, 151° 15.00 E), North Stradbroke Island (27° 20.50 S, 153° 33.73 E), Yongala (19° 18.51 S, 147° 37.10 E) and Darwin Harbour (DAR; 12° 24.00 S, 130° 46.08 E) span ~30° of latitude along the Australian continental shelf, occupying temperate, subtropical and tropical waters with minimum and maximum sea surface temperatures (SST) ranging between 11 °C and 31 °C. They are positioned to maximize spatial representation of Australian marine bioregions (as defined in Richardson et al 2020). Phylogenetic molecular markers were resolved to the level of sequences with single nucleotide differences, or amplicon sequence variants (ASVs). We defined optimal performance based on relative abundance in relation to environmental temperature, resulting in a species temperature index (STI) for each ASV.”

Line 122: Statistical tests and values are not fully reported in these results, such as varied CTI ~ temperature relationship.”

We now provide all parameters from linear regression between CTI ~ temperature for all kingdoms in Supplementary Data Table 3 including the global relationship, the relationship for voyage data at subsets therein (<10C, 10-20C and >20 C to show the changing slope of the relationship) and at each National Reference Station. Relationships between other environmental variables and community weighted indices are provided in Supplementary Table S5.

Line 348: These results don't seem to show up in pictures or tables.

As part of the revised Figure 6 and Supplementary Figures 26/27/28 we now provide heatmaps detailing the predominant taxa (at genus level for Bacteria and Archaea, the ~Family level for Eukaryotes and the ~Class level for Chloroplasts) that are selected for or against during each month of the heatwave.

Line 388: Additional evidence is needed to support these conclusions.

Statistical test results for each kingdom are provided in Supplementary Table 6-8.

Line 452: The authors do not report the sequence of the sample range, and prove that the depth of the sample sequencing is sufficient. Whether the microbial data is flattened or not, the result of relative abundance may affect the establishment of the model. If using the rarefied samples is needed to improve the robustness of the findings?

We concur with the reviewer that the outline of bioinformatics and statistical procedures was unclear and inadequate and have attempted to substantially address these issues

We now provide information on the initial number of sequences in each sample as well as the final numbers after QC measure in Supplementary Table 10.

We show that the community weighted indices are robust to differences in pre-processing bioinformatic strategy including rarefaction by comparing our results to those from the Australian Microbiome (AM).

The AM houses the same dataset but uses alternate ASV calling method based on the USEARCH algorithm (see the Australian Microbiome Methods manual https://confluence.csiro.au/display/ASM/meth_5.1+Amplicon+Analysis for details). Data is also rarefied to 20,000 reads per sample. We compare the community weighted indices output from both approaches in Supplementary Table 9 to show they are extremely highly correlated. The following text has been added to the methods section.

“To test the effect of upstream bioinformatics choices on the KD approach to community weighted index generation, we compared the results detailed here with those that are now output on an ongoing basis by the Australian Microbiome (AM). The AM generates the community weighted indices from the same sample set using

the same code as described here, but, for historical reasons, uses different intermediate bioinformatic protocols (see the AM methods manual for details https://confluence.csiro.au/display/ASM/meth_5.1+Amplicon+Analysis). The major differences between protocols are that the AM utilises the USEARCH algorithm (Edgar, 2010) to resolve ASVs (as opposed to DADA2 herein) and species and community indices are generated using samples rarefied to 20,000 reads (as opposed to un-rarefied sample with >5000 reads here). Because the KD approach relativises the input data as part of the code, these pre-processing protocol changes result in almost negligible difference in outcome (Supplementary Table 9). For example, linear model parameters correlating CTI estimates (using the KD approach) from here against the AM estimates for Bacteria (Estimate = 1.006, Std. Error = 0.0007, t-value = 1427.81, r2 = 0.998, F-statistic = 2.039e+06(1,3182), p < 2.2e-16), Archaea (Estimate = 0.998, Std. Error = 0.002, t-value = 594.87, r2 = 0.992, F-statistic = 3.539e+05(1,3034), p < 2.2e-16) and Eukaryotes (Estimate = 0.995, Std. Error = 0.002, t-value = 652.57, r2 = 0.994, F-statistic = 4.258e+05(1,2436), P<2.2e-16) are all highly significant. ”

Line 468: Whether this treatment that observations of zero abundance were replaced in ASV tables with a non-zero estimate value is reasonable, because the general ASV table usually has very high zero values.

Zero replacement tables were only used for nMDS and SIMPER analyses and not for generation of microbial indices, which used relative abundance tables - see next comment

Line 481: The authors calculated the optima and range of relationship of each ASV with available environmental parameters, however, whether there are false positives in low abundance, especially in ASVs converted from zero values. Whether it is more ecologically meaningful to retain only high abundance or core microorganisms for analysis.

We understand the reviewers concern here and now report in the methods that relative abundance tables were used for indices calculations and Zero replacement tables were used for nMDS and SIMPER analyses.

In terms of retaining only high abundance organisms, we note that the community weighted indices are, by nature, weighted by abundance so high abundance taxa have a proportionally greater impact on the value.

Also, ASVs were required to have at least 100 replicate estimates (from subsampling by replacement) for their values to be used to calculate community indices, so rare ASVs that did not satisfy this criteria were indeed excluded.

Line 508: Due to the accuracy of annotation and sequencing, different ASVs may represent the same species, which is different from the index constructed by plankton or other marine organisms at a more precise level of species classification. And whether the applicability in species results annotated by 16s or 18s sequencing is verified.

This is a very good point. As the reviewer correctly points out, it is generally unwise to precisely relate any individual rRNA ASV to a single eukaryotic “species”.

We have now tried to encapsulate some of this uncertainty in our language and state in the conclusions that

“While direct comparisons of molecular methods to data generated by other methods (e.g. microscopy) at the species level should be undertaken with caution (see Methods, Supplementary Methods and Supplementary Table 11), comparison between the magnitude and directional response of community level indices across taxonomic boundaries (e.g. microbes to phytoplankton to zooplankton to fish) will likely prove fruitful as generalised whole of ecosystem indicators.”

We have also made a considerable effort to place our species niche indices into more context to imbue confidence in their validity. We provide evidence they recapitulate well established patterns of marine microbial taxa in each kingdom, as well as do direct comparison where applicable to phytoplankton species temperature indices generated by traditional methods. The following section has been added to the Methods and a new section to the Supplementary materials titled “Examples of Baseline Data”.

“While independent verification of all indices is beyond the scope of this work, we provide working examples to highlight how our index dataset recapitulates well established baseline patterns of critical taxa in marine environments generated for each kingdom while supporting exploration and identification of novel findings. Examples describe niche characteristics of clades within the marine cyanobacteria (Supplementary Figure 32), the Marine Group II Archaea (Supplementary Figure 33) and the unicellular green algae of the Class Mamiellophyceae within the Eukaryotes (Supplementary Figure 34). Finally, to highlight the discovery aspect of the data we present the niche parameters of largely uncharacterised and uncultivated dinoflagellates in clades I-IV within the Syndiniales, a ubiquitous group of endoparasites. Our results highlight how these clades are structured along oceanic gradients, with some displaying very tight ecological constraints that could be used to track environmental change.

We also provide a comparisons of the STI derived here with those estimated for some phytoplankton species that are common along the eastern Australian coast as estimated by light microscopy based on whole water and continuous plankton recorder dataset by Ajani et al (2020) (Supplementary Table 11). These comparison highlight both synergies and discrepancies between the approaches and suggest caution should be employed when directly comparing the two (Supplementary methods, Supplementary Table 11).“

Line 538: If possible, making the code available would help to assess the statistical rigor and validity of the analysis. Furthermore, this will allow future scientists to build upon the similar work.

The code, along with all molecular and environmental data, is publicly available and we have added the following section to the manuscript to reflect this.

“Code Availability

All RCode required for the calculation of microbial species and community indices using the kernel density approach from a 3 column species x abundance x environmental-variable table is publicly available on the Australian Microbiome Github (https://github.com/AusMicrobiome/microbial_ocean_atlas).”

Reviewer #2 (Remarks to the Author):

The COMMSBIO-23-2150-T study describes how a marine heat wave causes significant shifts in pelagic microbiology. The topic is highly relevant and timely as climate change threatens marine ecosystems and organisms worldwide. Marine heatwaves (MHWs) are prolonged anomalous ocean warming events that have attracted attention as triggers of, for example, coral bleaching events. Linked to global warming, MHWs have been increasing in frequency, intensity, and duration. Yet, most of the research to date has focused on oceanography or macroorganisms. Investigations into the relationship between microorganisms and climate change have long been hampered by comprehensive data sets and methodological limitations. Furthermore, most marine microbial time series that are able to detect trends over time are located in the northern hemisphere, leaving huge knowledge gaps in the southern ocean regions. The present study not only uses an impressive dataset of thousands of samples to fill the knowledge gaps on the impact of MHWs on marine microbes in the Southern Hemisphere, but also generates novel indices and important insights that are transferable to other ocean regions, extreme events and groups of organisms. I therefore think that this paper is highly relevant and of interest for the microbial community and a wider field as well. The sequences and environmental data were found in the respective databases and the methods were described in a comprehensive way such that the data analysis should be reproducible.

Despite the overall great impression, I have a few comments that should be addressed:

I am confused by two sets of supplementary figures and tables (Extended data figures and Supplementary figures). I recommend combining these files into one document.

We have now combined all extra material into one Supplementary Data product.

Fig2 - legends and formatting, the figure resolution in the provided pdf was bad. Please provide a better resolution and larger captions and legends.

We have remade Figure 2 to hopefully enhance the readers comprehension from this image and have moved some of the previous images to supplementary materials..

Fig3 – please provide axis captions

We have remade Figure 3.

Extended Data Figure 7. This figure is very striking and describing overall biogeographic community/niche patterns. Together with my below comment on bacteria community composition (L346ff) this would make an excellent overview figure.

We agree with the reviewer that a greater focus on the basin scale biogeographic patterns presented in the supplementary materials would be of interest, but feel it is out of scope of this paper where we aim to keep the focus on marine heatwaves.

L346ff - which taxa have the specific niches and can adapt most to the heatwave -> extend the presentation of results and discussions related to the identity of “winners” and “losers”, also did alpha diversity or community evenness differ during MHW event and between bacteria, archaea, and eukaryotes? Please include a figure representing the community composition of communities in the different clusters or possibly in combination with Fig.3.

We thank the reviewer for the suggestions and have incorporated alpha diversity indices into the main paper. Figure 6 and Supplementary Figure 26/27 now present an analysis of how the strongly seasonal patterns in Shannon and Inverse Simpson diversity estimates are affected by MHW conditions. Mention of this finding has also been included in the Abstract. Diversity indices are now presented along with a heatmap representation of the taxa that contribute most to the difference between MHW and non-MHW samples (i.e. “winners and losers”) and extra text has been added to the discussion around these images. Heatmaps and diversity indices are now combined with the previous analysis identifying “uncommon” organisms under a new heading “Marine heatwaves alter the diversity, structure and composition of microbial assemblages.”

L360 - It would be interesting to see if/how the authors think that the study relates to the Metabolic Theory of Ecology (MTE) or emergent principles regarding increasing temperatures - see for example the recent review by Gralka 2023 <https://doi.org/10.1093/icb/icad060>.

We thanks the reviewer for the suggestion to think about our results under these frameworks. By comparison to the few other works examining MHW effects on microbial assemblages have made some suggestions as to potentially emergent characteristics of warming temperate ecosystems in the abstract, discussion and conclusions. E.g. Line 381

“Taken together, these common responses (i.e. MHW phytoplankton assemblages dominated by small, unicellular chlorophytes, dinoflagellates and cyanobacteria, along with increased numbers of photoheterotrophs), may be fundamental results of warming in temperate marine waters. Importantly, these shifts can have profound ecosystem consequences as the rates, magnitude and fate of carbon fixed by small unicellular phytoplankton can be quite different to that fixed by lager organisms such as diatoms (Messer et al 2020).“

L378 – “introduction of new species into the environment” Where would the species be introduced from if not transported with currents or from the rare biosphere?

We have removed this sentence as part of restructuring the paper.

L388-392 – is this trend statistically significant?

We now provide results of statistical tests for diversity and “proportion of uncommon taxa ” in Supplementary Tables 6-8

L446 and elsewhere, refer to the full amplicon identifier 16S rRNA instead of just “16S”. Which primers/gene regions were used?

We now refer to the full amplicon identifier for all assays and have added the primer sequences to the methods.

L467 was one combined ASV table across stations/projects generated for downstream analyses? How many samples did not pass the filter thresholds? A table with sequencing results and #sequences passing the bioinformatic filters should be added to the supplementary files.

Yes, one combined table was used for the analyses. We have added a table detailing the numbers of sequences in each sample before and after QC as Supplementary Table S10.

L490 – It would be useful for the less experienced reader if the authors could show some examples of calculated species ranges as supplementary figure, e.g. of the species that were found to be least/best adapted to higher temperatures.

We now display some examples of raw data, kernel density model and final model output parameters for three “winners” and three “losers” under a new section in the Supplementary Materials titled “Examples of the Kernel Density Estimation approach to niche characterisation” including Supplementary Figure 31 and supplementary Table 12.

L507f – Please add the mathematical formulas for each index.

This has been done.

REVIEWERS' COMMENTS:

Reviewer #1 (Remarks to the Author):

The MS was well revised according to the review comments and can be accepted now.

Reviewer #2 (Remarks to the Author):

The authors have addressed all of my previous comments satisfactory and I think the manuscript now reads very well.